# Cognitive impairment in Chagas disease patients in Brazil, 2007–2021: A cross-sectional study

Carla J. Serrano[1,2], Maria E. Lisbôa-Marques[1], Thiago Cerqueira-Silva[1,3],
Leila S. B. Santos[1], Murilo A. Oliveira[1], Iuri Ferreira Felix[1], Paulo R.S.P. de Sousa[1],
Leonardo G.M. Cardoso[1], Pedro J.R. Muiños[1], Renata M. Maia[1], Marília B. Catto[1],
Eric Aguiar Wittlich[1], Lucy Rodrigues-Ribeiro[1], Victor L.P.P. Botelho[1],
Maria Carmo P. Nunes[4], Antonio Luiz P. Ribeiro[4], Lucas C. Barbosa e Silva[4],
Roque Aras[1], Karen L. Furie[5], Jamary Oliveira Filho[1,2]*

**1** Stroke and Cardiomyopathy Clinics, Hospital Universitário Professor Edgard Santos, Universidade Federal da Bahia, Salvador, Brazil, **2** Programa de Pós-Graduação em Ciências da Saúde, Faculdade de Medicina da Bahia, Universidade Federal da Bahia, Salvador, Brazil, **3** Faculty of Epidemiology and Population Health, London School of Hygiene & Tropical Medicine, London, United Kingdom, **4** Department of Internal Medicine, Faculdade de Medicina, and Telehealth Center and Cardiology Service, Hospital das Clínicas, Universidade Federal de Minas Gerais, Belo Horizonte, Brazil, **5** Neurology Service, Brown University, Providence, Rhode Island, United States of America

* jamary@mail.harvard.edu

## Abstract

### Introduction

Chagas Disease (CD) is frequently associated with heart failure (HF). Cognitive impairment is reported, but whether it results from CD or is a nonspecific symptom of HF is unknown. We aimed to compare cognitive function of HF patients with or without CD.

### Methods

Multicenter cross-sectional study of HF patients. Investigators blinded to the etiology of HF evaluated global cognition and domains of memory, executive and visuospatial function. Logistic regression tested the association between CD and cognitive impairment (Z-score < −1.5) in each domain adjusted for age, sex, educational level and left ventricular ejection fraction.

### Results

We recruited 518 patients, 250 (48.3%) with CD. Cognitive impairment was more common in CD vs. non-CD patients (27.1% vs 13.1%, p < 0.001), mostly in memory (10.4% vs 5.0%, p = 0.022) and visuospatial function (45.2% vs 29.6%, p < 0.001). In the multivariable analysis, CD remained associated with global cognitive impairment (odds ratio 1.90; 95% CI 1.13–3.21, p = 0.016) and visuospatial function impairment (OR 1.56; 95% CI 1.02–2.39, p = 0.039).

**Data availability statement:** All de-identified data used to replicate the study's findings are available upon request to the Institutional Ethics Committee (email: cep.hupes@ebserh. gov.br). Any data can only be shared with ethical approval to ensure it is shared in accordance with participant consent, as the data contain potentially sensitive personal health information.

**Funding:** This work was supported by the National Institutes of Health, grant number RO1NS064905 to JOF and by the Brazilian National Scientific Council (CNPq), grant number NCT01650792 to JOF and grant number 310790/2021-2 to ALPR, and by the Royal Society, grant number NIF\RI\231435 to TCS, and by Minas Gerais State Agency for Research and Development (FAPEMIG), grant number RED 00192-32 to ALPR. The funders had no role in study design, data collection and analysis, decision to publish, or preparation of the manuscript.

**Competing interests:** The authors have declared that no competing interests exist.

## Discussion

Chagas disease is associated with cognitive impairment independently of heart failure severity, suggesting other competing mechanisms.

## Author summary

Heart failure is the most well-known manifestation of Chagas disease, caused by the protozoan *Trypanosoma cruzi.* Less studied are symptoms attributed to the central nervous system. Previous studies have shown some degree of cognitive impairment in individuals affected by Chagas disease. It remains unclear whether the disease itself directly causes the cognitive deficits observed in patients with Chagas disease or if they result from heart failure, as it can lead to reduced blood flow to the brain and inflammation, contributing to cognitive decline. To better understand this relationship, we studied the cognitive function of 518 patients diagnosed with heart failure, both with and without Chagas disease, assessing memory, as well as executive and visuospatial abilities. Through this study, we concluded that cognitive impairment was more common in individuals with Chagas disease, particularly in aspects related to memory and visuospatial abilities. Furthermore, the relationship between Chagas disease and cognitive impairment was independent of the presence or severity of heart failure, suggesting that other competing mechanisms such as chronic inflammation may be present.

## Introduction

Chagas Disease (CD) is a neglected infectious disease, first described in 1909 by Carlos Chagas [1]. Approximately 30% of the infected patients will develop at least one of the chronic forms of the disease. Chagas cardiomyopathy is the most severe clinical manifestation of the CD's spectrum and remains an important burden for public health in Latin-America [2,3].

Brain involvement in CD is usually attributed to the indirect effect of cardiomyopathy, by embolism or hypoperfusion, but brain atrophy has been shown to be independent of cardiac disease severity [4–6]. Clinical repercussion of these anatomical findings is unknown and confounded by the effects of heart failure (HF) on cognition [7]. Additionally, it is unknown whether clinical findings are residual deficits from the acute phase meningoencephalitis or if there is active infection-related cognitive decline from neurodegeneration [8,9]. Indeed, in a pre-clinical model of chronic Chagas disease, behavioral and cognitive changes were dissociated from cardiac involvement and associated with parasite load in the brain, interferon-gamma, tumor necrosis factor and serotonin; and partially reversed by treatment with fluoxetine, a serotonin-uptake inhibitor [10].

Cognitive functions can be characterized as a set of mental processes that enable the brain to process information and acquire knowledge. The main domains that constitute

cognition are intelligence, attention, reasoning, perception, language, memory, executive functions, and visuospatial functions [11,12]. Not until 1994 the first study describing the impact of CD in cognition was published in Argentina [13], followed by two others in endemic regions of Brazil [14,15]. Such studies demonstrated low global scores; and abnormalities in many cognitive domains, suggesting an association between chagasic cardiomyopathy and some degree of cognitive dysfunction [13–15]. However, authors could not exclude residual confounding from ventricular dysfunction itself or cerebrovascular disease.

In face of the possible cognitive impairment caused by CD, the present study aims to determine in HF patients if an association between CD and cognitive function is present when compared to non-CD patients; and to discriminate which cognitive domains are primarily affected in CD.

## Methods

### Ethics statement

The present study was approved by the Brazilian Nacional Ethics Committee (CONEP) under registry number 529.140 and is in accordance with the STROBE statement. All patients signed an informed consent form.

### Study population

We performed a cross-sectional study, between 2007 and 2021, using a cohort of patients with HF from four Brazilian hospitals, including the Professor Edgard Santos University Hospital, Santa Izabel Hospital, Ana Nery Hospital, located in Salvador, Bahia, and the Bias Fortes Hospital, from the Federal University of Minas Gerais. Sample size and inclusion/exclusion criteria were previously defined [16]. Subjects who met inclusion criteria were older than 18 years and had a clinical diagnosis of HF, defined according to Framingham criteria (any previous history of two major criteria; or a combination of one major and two minor criteria) [17]. New York Heart Association functional class was used to classify current HF status. Transthoracic echocardiography data was collected including quantification of left ventricular function (left ventricular ejection fraction). Stroke history was excluded based on screening for previous stroke symptoms by the Questionnaire for Verifying Stroke-Free Status (QVSFS) [18]. Further exclusion criteria were history of any other neurodegenerative disease (including dementia), use of anticoagulants and subjects unable to provide legal consent.

### Clinical and cognitive assessment

We collected all information by consulting medical records and confirmed collected data using a standardized questionnaire (sex, age, residence, place of birth, education level, diagnosis of cardiomyopathy, physical examination, electrocardiogram, transthoracic echocardiogram, quality of life and risk factors). Chagas disease was defined by the presence of a positive serologic test on chart review (hemagglutination or immunofluorescence test), confirmed by an ELISA assay performed at a single central laboratory (Hospital Universitario Professor Edgard Santos Immunology Laboratory). On the same day of recruitment, a second investigator blinded to all clinical data applied validated Portuguese versions of the following tests: Mini-Mental State Exam (MMSE) as a global cognition test [19]; the Brief Cognitive Screening Battery (BCSB), which evaluates a series of immediate and delayed visual memory tests, semantic verbal fluency (maximum number of animals named in one minute) and also includes the clock-drawing test [20]; the digit span subtest from Wechsler Intelligence Scale (WAIS), which aims to measure specifically attention and working memory [21–23]; the Rey Complex Figure Test (RCFT), which evaluates visuo-constructive ability, visuospatial function, planning strategies and visual memory [24]; and Luria's fist-edge-palm test (FEPT), which evaluates praxis and executive function [25].

### Cognitive analysis

Scores of each cognitive test were adjusted for age and transformed into a Z-score based on normalized values of the Brazilian population [21,24–26], except for the latest recall of Rey Complex Figure Test, which was based on the United States of America population [27].

Cognitive tests were grouped into three cognitive domains: memory (recall of RCFT [24]; WAIS digit span subtest [21–23] – direct order; incidental, immediate, late recall and recognition memory [20]); executive function (verbal fluency – animals [20]; FEPT and WAIS digit span subtest – inverse order); and visuospatial function (clock-drawing test [20] and copy of RCFT [24]). Global cognitive score was the mean value of these 3 domains. We defined cognitive impairment in each domain as a Z-score < −1.5.

## Statistical analysis

Statistical analysis was performed through R Software version 4.0.4 (R Core Team) [28]. Numerical variables were described as mean and standard deviation when normally distributed and median and interquartile range when non-normally distributed, judged based on a Shapiro-Wilk test. Absolute and relative (%) numbers described categorical variables. Baseline characteristics were compared using t test, Mann-Whitney test or Fisher exact test, as appropriate. Missing values variables in the final models were imputed using multiple imputation by chained equations using random forest algorithm [29]. Logistic regression models were built for each cognitive domain, adjusted for sex, age, educational level and left ventricular ejection fraction (LVEF), to detect the independent effect of Chagas cardiomyopathy on impaired cognition. *p*-values were two-tailed, with values < 0.05 considered significant.

## Results

Of 1,000 charts evaluated, 518 patients met inclusion criteria and did not present any exclusion criteria. Among them, 250 (48.2%) had CD. In this cohort, 160 patients with CD had heart failure, with functional classes ranging from II to IV, compared to 188 with other cardiomyopathies (OC). Cardiomyopathy etiologies in the OC group were hypertensive (28%), ischemic (22%), idiopathic dilated (19%), post-viral myocarditis (5%), alcoholic (4%) or other (22%). Functional class II heart failure was more prevalent in the groups, 45.4% among those with CD and 46.4% among those with OC. The distribution of city of birth size was similar in both groups: small cities (<50000 inhabitants) 39% in Chagas vs 52% non-Chagas patients; medium-sized cities (50000–100000 inhabitants) 10% in Chagas and 15% in non-Chagas patients; and large (>100000 inhabitants) 51% in Chagas and 33% in non-Chagas subjects. Recruitment occurred mostly before the COVID19 pandemic – 492 (95.7%) subjects, with a similar distribution between Chagas [240 (97.2%)] and non-Chagas [252 (94.4%)] patients (p = 0.132).

Chagasic group was older (57 ± 11 vs 54 ± 14 years), more likely to be female and presented with a lower educational level, with 26.2% of illiterate individuals, compared to 9.5% in the non-chagasic group. In our sample, chagasic patients less often had a previous history of myocardial infarction (MI), coronary artery disease (CAD) and alcohol use; they also had a better left ventricular ejection fraction (LVEF) (median 53.9% [35–68] vs 37% [28–50]), indicating less severe ventricular dysfunction (Table 1).

Chagas disease was associated with a lower score on MMSE when compared to non-CD patients (median Z-score 0.193 [−0.975 to 0.589] vs 0.328 [0.453 to 0.850]. Global cognitive impairment was more frequent in CD vs non-CD patients (27.1% vs 13.1%, p < 0.001). When stratified by cognitive domains, both memory and visuospatial function domains were more frequently impaired in CD vs non-CD patients (Table 2).

In the multivariable analysis adjusted for age, sex, educational level and left ventricle ejection fraction (Table 3), CD remained associated with global cognitive impairment (OR 1.90; 95% CI 1.13–3.21, p = 0.016) and visuospatial function impairment (OR 1.56; 95% CI 1.02–2.39, p = 0.039). While memory impairment did not remain associated with CD. Additional adjustment for alcohol use or cardiac rhythm did not change overall results.

## Discussion

In this large cohort of HF patients, our main finding was an independent association between CD and impaired cognition. Although a few studies have investigated this topic [13–15], ours is the first to consider simultaneously several

PLOS
Neglected Tropical Diseases

**Table 1. Clinical, social and demographic characteristics of 518 patients with Chagas disease cardiomyopathy (CDC) and other cardiomyopathies (OC).**

| Variables | CDC (n = 250) | OC (n = 268) | P-value |
|---|---|---|---|
| **Female sex, n (%)** | 162 (64.8) | 123 (45.9) | <0.001 |
| **Age, mean (SD)** | 57 (11) | 54 (14) | 0.015 |
| **Missing, n** | 6 | 4 | |
| **Educational Level, n/N (%)** | | | |
| Illiteracy | 64 (26.2) | 25 (9.5) | <0.001 |
| Incomplete elementary school | 137 (56.1) | 127 (48.5) | |
| Complete elementary school | 16 (6.6) | 32 (12.2) | |
| Incomplete high school | 12 (4.9) | 19 (7.3) | |
| Complete high school | 15 (6.1) | 49 (18.7) | |
| University education | 0 (0.0) | 10 (3.8) | |
| **Missing, n** | 6 | 6 | |
| **Ethnicity, n (%)** | | | |
| Afro-descendant | 204 (85.7) | 220 (85.6) | 0.085 |
| White | 26 (10.9) | 35 (13.6) | |
| Other | 8 (3.4) | 2 (0.8) | |
| **Missing, n** | 12 | 11 | |
| **BMI (kg/m²)** | 25.4 (7.8) | 26.0 (8.0) | 0.525 |
| **Functional class (NYHA), n (%)** | | | |
| I | 58 (26.6) | 51 (21.3) | 0.662 |
| II | 99 (45.4) | 111 (46.4) | |
| III | 49 (22.5) | 56 (23.4) | |
| IV | 12 (5.5) | 21 (8.8) | |
| **Missing, n** | 32 | 29 | |
| **LVEF (%), median (IQR)** | 53.92 (35.2 - 68.2) | 37.1 (28.0 - 50.42) | < 0.001 |
| **Missing, n** | 56 | 46 | |
| **Rhythm on EKG, n (%)** | | | 0.3 |
| Sinus | 180 (90) | 166 (94) | |
| Atrial fibrillation | 17 (8.5) | 8 (4.5) | |
| Others | 3 (1.5) | 2 (1.2) | |
| **Missing, n** | 50 | 92 | |
| **Conduction abnormalities (%)** | | | |
| Atrioventricular block | 24 (12) | 10 (5.7) | 0.15 |
| **Missing, n** | 51 | 92 | < 0.001 |
| Right branch block | 82 (41) | 21 (12) | <0.001 |
| **Missing, n** | 51 | 95 | |
| Left branch block | 18 (9.2) | 43 (25) | |
| **Missing, n** | 55 | 93 | |
| **Comorbidities, n (%)** | | 73 (28.9) | < 0.001 |
| CAD | 31 (13.2) | 15 | 0.334 |
| **Missing, n** | 16 | 62 (23.7) | 0.185 |
| Diabetes | 49 (19.9) | 6 | < 0.001 |
| **Missing, n** | 4 | 182 (69.7) | |
| Hypertension | 156 (63.9) | 49 (18.7) | |
| **Missing, n** | 20 (8.3) | | |

*(Continued)*

**Table 1.** (Continued)

| Variables | CDC (n = 250) | OC (n = 268) | P-value |
|---|---|---|---|
| **Myocardial infarction** | | | |
| **Smoking, n (%)** | | | |
| **Never** | 135 (56.7) | 135 (51.3) | 0.236 |
| **In the past** | 99 (41.6) | 118 (44.9) | |
| **Current** | 4 (1.7) | 10 (3.8) | |
| **Missing, n** | 12 | 5 | |
| **Alcohol use, n (%)** | | | |
| **Never** | 66 (27.7) | 51 (19.5) | 0.012 |
| **Often – Past** | 47 (19.7) | 83 (31.7) | |
| **Ocasional – Past** | 78 (32.8) | 88 (33.6) | |
| **Occasional – Current** | 45 (18.9) | 37 (14.1) | |
| **Often – Current** | 2 (0.8) | 3 (1.1) | |
| **Missing, n** | 12 | 6 | |
| **COVID-19, n (%)** | | | |
| **Before** | 240 (97.2) | 252 (94.4) | |
| **Missing, n** | 3 | 1 | |

NYHA *New York Heart Association;* **LVEF:** *Left Ventricle Ejection Fraction;* **CAD:** *Coronary Artery Disease;* **EKG:** *Eletrocardiogram*

**Table 2. Cognitive impairment of 518 patients with Chagas disease (CD) cardiomyopathy and non-CD cardiomyopathies.**

| Variables | CD (n = 250) | No CD (n = 268) | P-value |
|---|---|---|---|
| **Memory abnormality, n (%)** <br> Missing, n | 25 (10.4) <br> 10 | 13 (5.0) <br> 8 | 0.022 |
| **Executive function abnormality, n (%)** <br> Missing, n | 78 (32.5) <br> 10 | 84 (32.6) <br> 10 | 0.989 |
| **Visuospatial function abnormality, n (%)** <br> Missing, n | 103 (45.2) <br> 22 | 75 (29.6) <br> 15 | <0.001 |
| **Global cognition abnormality, n (%)** <br> Missing, n | 65 (27.1) <br> 10 | 34 (13.1) <br> 8 | <0.001 |

**Table 3. Effect of Chagas disease on global cognitive impairment and in each cognitive domain.**

| Variables | Odds ratio* | 95% CI | P-value |
|---|---|---|---|
| **Memory** | 1.74 | 0.76–3.99 | 0.200 |
| **Executive function** | 0.73 | 0.47–1.13 | 0.200 |
| **Visuoespatial function** | 1.56 | 1.02–2.39 | 0.039 |
| **Global cognition** | 1.90 | 1.13–3.21 | 0.016 |

*Adjusted for: age, sex, educational level and left ventricle ejection fraction. **CI:** *confidence interval.*

important confounders of cognitive impairment, such as age, educational level and cardiac disease severity. The first study to evaluate cognition in CD evaluated patients with the cardiac form of CD, compared with controls without CD or cardiac disease [13]. The second study was a population-based study in an endemic area, used only the MMSE test

and adjusted their analyses by the presence of electrocardiographic abnormalities [14]. While both studies show an association between CD and impaired cognition, it is likely that significant confounding from cardiac disease severity existed, as an estimated 43% of patients with HF will develop cognitive impairment over time [7] In contrast, in our study mild imbalances found between groups in age, sex, left ventricular function and educational level were adjusted for in all statistical analyses.

Our group previously studied cognition in CD in a smaller sample of 79 patients [15], in which 37 patients with CD were compared with 42 non-CD patients, showing an independent association between CD and only one visuospatial function test. The present study confirms our preliminary findings in a larger sample and allowed for more robust multivariable adjustment.

Another finding of our study was a detailed description of which cognitive domains are more severely affected in CD. The study by Mangone et al. showed impaired cognition in attention, executive function and memory, but compared CD patients to controls without cardiac disease [13]. The other study reported cognitive impairment in one visuospatial function test, which confirms our present findings [15]. This rather unique pattern of domain-specific impairment is non-overlapping with early Alzheimer-type impairment, which mostly affects memory; or vascular cognitive impairment, which more frequently affects executive function [30].

The mechanisms underlying the pattern of CD-associated cognitive impairment are under investigation in another study by our group [16]. Stroke is more common in CD when compared to other cardiomyopathies [5,31,32] and brain infarcts have been described in 18% of autopsied patients with CD cardiomyopathy [33]. Although we excluded patients with known history of stroke, silent infarcts could partly contribute to the burden of cognitive deficits. Brain atrophy is another potential mechanism for cognitive impairment and has been reported more frequently in head CT of patients with CD when compared to non-CD controls [4]. Inflammatory biomarkers, seen in Chagas disease [34], have been associated with brain atrophy in a cohort of 1926 individuals [35]. In an experimental model of chronic Chagas disease, inflammatory biomarkers were also associated with cognitive and behavioral abnormalities [10]. Recently, another infectious disease, COVID-19, has been associated with brain atrophy and cognitive impairment in a serial MRI study [36]. We can only speculate that some or all these mechanisms may justify our clinical findings in this study.

Little is still known about CD impact on cognitive function. Our multicenter study is the largest sample to date with detailed cognitive assessment of CD patients and strict exclusion criteria, which were previously linked to affect cognitive function, such as neurodegenerative and cerebrovascular diseases. As occurs with any cross-sectional design, the possibility of residual confounding not covered by the current adjustment may remain as a limitation. It is important to emphasize that our study was initiated prior to the pandemic period, and for this reason, patients who were infected with COVID-19 were not excluded from the study. However, most (95.7%) patients were recruited before the pandemic and the distribution of patients recruited before and after the pandemic were not different between Chagas and non-Chagas patients.

We did not include a non-diseased control group, because we wanted to control for the presence of cardiac disease and investigate the isolated effect of CD on cognition. A neuroimaging component of the present study is ongoing and should help answering which mechanisms are implicated in CD-associated cognitive impairment. Finally, low socioeconomic status is a significant factor leading to cognitive and behavioral dysfunction. Both groups in the present study came from study sites seeing patients from the public health system in Brazil, which oversees mostly patients with low socioeconomic status. Since educational status closely correlates with socioeconomic status, we believe that our analyses adjusted for educational status partially accounted for this important factor.

Our research team is composed of physicians, medical students, and a psychologist. The multidisciplinary nature of the research proposes a collaboration between neurology, cardiology, and neuropsychology, providing patients with not only medical care but also a space for support, emotional assistance, and psychoeducation in the face of the cognitive difficulties they experience.

## Author contributions

**Conceptualization:** Karen L Furie, Jamary Oliveira Filho.

**Data curation:** Thiago Cerqueira-Silva, Eric Aguiar Wittlich, Jamary Oliveira Filho.

**Formal analysis:** Thiago Cerqueira-Silva.

**Funding acquisition:** Karen L Furie, Jamary Oliveira Filho.

**Investigation:** Carla Serrano, Maria Eduarda Lisbôa-Marques, Leila Silva Brito Santos, Murilo Araújo Oliveira, Iuri Ferreira Felix, Paulo Roberto Sampaio Peixoto de Sousa, Leonardo Galvão Machado Cardoso, Pedro José Ramiro Muiños, Renata Martins Maia, Marília Bazzo Catto, Eric Aguiar Wittlich, Lucy Rodrigues Ribeiro, Victor Luis Peixoto Pereira Botelho, Maria C P Nunes, Antonio Luiz P Ribeiro, Lucas C. Barbosa e Silva, Roque Aras.

**Methodology:** Carla Serrano, Maria Eduarda Lisbôa-Marques, Maria C P Nunes, Antonio Luiz P Ribeiro, Lucas C. Barbosa e Silva, Roque Aras, Jamary Oliveira Filho.

**Project administration:** Jamary Oliveira Filho.

**Supervision:** Maria C P Nunes, Antonio Luiz P Ribeiro, Roque Aras, Jamary Oliveira Filho.

**Validation:** Antonio Luiz P Ribeiro, Lucas C. Barbosa e Silva, Jamary Oliveira Filho.

**Visualization:** Jamary Oliveira Filho.

**Writing – original draft:** Carla Serrano, Maria Eduarda Lisbôa-Marques, Karen L Furie, Jamary Oliveira Filho.

**Writing – review & editing:** Carla Serrano, Maria Eduarda Lisbôa-Marques, Thiago Cerqueira-Silva, Leila Silva Brito Santos, Murilo Araújo Oliveira, Iuri Ferreira Felix, Paulo Roberto Sampaio Peixoto de Sousa, Leonardo Galvão Machado Cardoso, Pedro José Ramiro Muiños, Renata Martins Maia, Marília Bazzo Catto, Eric Aguiar Wittlich, Lucy Rodrigues Ribeiro, Victor Luis Peixoto Pereira Botelho, Maria C P Nunes, Antonio Luiz P Ribeiro, Lucas C. Barbosa e Silva, Roque Aras, Karen L Furie, Jamary Oliveira Filho.

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
