## [Decision Letter · Decision Letter 0]

30 Oct 2024

PNTD-D-24-01215CHAGAS DISEASE-ASSOCIATED COGNITIVE IMPAIRMENTPLOS Neglected Tropical Diseases Dear Dr. Serrano, Thank you for submitting your manuscript to PLOS Neglected Tropical Diseases. After careful consideration, we feel that it has merit but does not fully meet PLOS Neglected Tropical Diseases's publication criteria as it currently stands. Therefore, we invite you to submit a revised version of the manuscript that addresses the points raised during the review process. Please submit your revised manuscript within 60 days Dec 29 2024 11:59PM. If you will need more time than this to complete your revisions, please reply to this message or contact the journal office at plosntds@plos.org. Please include the following items when submitting your revised manuscript:* A rebuttal letter that responds to each point raised by the editor and reviewer(s). You should upload this letter as a separate file labeled 'Response to Reviewers '. This file does not need to include responses to any formatting updates and technical items listed in the 'Journal Requirements' section below.* A marked-up copy of your manuscript that highlights changes made to the original version. You should upload this as a separate file labeled 'Revised Manuscript with Track Changes '.* An unmarked version of your revised paper without tracked changes. You should upload this as a separate file labeled 'Manuscript '. If you would like to make changes to your financial disclosure, competing interests statement, or data availability statement, please make these updates within the submission form at the time of resubmission. Guidelines for resubmitting your figure files are available below the reviewer comments at the end of this letter. We look forward to receiving your revised manuscript. Kind regards, Hira L Nakhasi, Ph.D.Section EditorPLOS Neglected Tropical Diseases Hira NakhasiSection EditorPLOS Neglected Tropical Diseases

Shaden Kamhawi

co-Editor-in-Chief

Paul Brindley

co-Editor-in-Chief

 **Journal Requirements:** **Additional Editor Comments (if provided):****Reviewers' Comments:** Reviewer's Responses to Questions

**Key Review Criteria Required for Acceptance?**

**Methods**

-Are the objectives of the study clearly articulated with a clear testable hypothesis stated?

-Is the study design appropriate to address the stated objectives?

-Is the population clearly described and appropriate for the hypothesis being tested?

-Is the sample size sufficient to ensure adequate power to address the hypothesis being tested?

-Were correct statistical analysis used to support conclusions?

-Are there concerns about ethical or regulatory requirements being met?

Reviewer #1: see comments

Reviewer #2: The objective of the study is clearly articulated

The study design is appropriate, yet it will be important to provide additional information (personal or from other sources) in relation to the place of birth and time living in rural areas or in towns, of the patients analyzed in this study.

The sample size is sufficient. The statistical analysis is correct.

There are no ethical concerns

**Results**

-Does the analysis presented match the analysis plan?

-Are the results clearly and completely presented?

-Are the figures (Tables, Images) of sufficient quality for clarity?

Reviewer #1: see comments

Reviewer #2: Yes, the analysis presented matches the analysis plan,

the results are clearly presented, and the tables are clear.

**Conclusions**

-Are the conclusions supported by the data presented?

-Are the limitations of analysis clearly described?

-Do the authors discuss how these data can be helpful to advance our understanding of the topic under study?

-Is public health relevance addressed?

Reviewer #1: See comments

Reviewer #2: Yes, but as i will write below, I think the paper will gain comprehension if a small description of the patients origin (see later) is described.

Public health was not addressed

**Editorial and Data Presentation Modifications?**

Reviewer #1: See comments

Reviewer #2: The main question that i consider will add relevance to the manuscript is to indicate: 1) the place the persons (both those with Chagas and those without Chagas) were born (if at the country site, at small villages, or at big villages and towns). 2) if opportune, for how long these persons lived in rural areas.

**Summary and General Comments**

Reviewer #1: Prologs

The team assembled in this multicenter study is formed by respected researchers in the field of Chagas disease, particularly in areas such as cardiac diseases and neurological and behavioral disorders, publishing their findings in the dominant language, that is, English. Therefore, it was not a pleasant surprise to receive this MS submitted to PLoSNTD with several texts in Portuguese. The authors and, mainly, the editorial board of PLoSNTD should show some respect to all reviewers. It is unacceptable considering the high level of this scientific journal. Can I submit an article in a native Brazilian language (such as Tupi-Guarani) or in any Arabic or Asian language? If so, please inform in “Instructions to authors”. Furthermore, if the authors wish to restrict access to their data, please submit it to a Brazilian or Portuguese scientific journal. I am sure they will find some good ones.

MS revision:

The topic of this MS is quite relevant. A possible chronic nervous form of Chagas disease based on behavioral/cognitive changes (not in neurological alterations) has been neglected and is not currently recognized. Therefore, patients with Chagas disease may miss out on adequate care. In the last decades, several articles (clinical and preclinical) have addressed this issue. Some of these original precedent publications deserve to be acknowledged, as they paved the way for the present study.

The multicenter study included a significant number of patients, and the results are quite interesting. Some points remain elusive.

To facilitate the indication of revision, the lines in the revised version should be numbered.

Abstract:

Based on results in the literature, it is better to say, “Chagas disease is frequently associated with heart failure.”

The goals are clearly described (achieved?).

It is not clear what is a "whether from CD or nonspecific symptom of HF”. Please, clarify or change it.

It is unclear what " whether from CD or nonspecific symptom of HF" is. Please clarify or amend it.

The last sentence shows “…heart failure severity…”. Thus, the studied groups should be better described, including means of age, levels of cardiopathy, clinical exams, symptoms and signs used for classification HF, register of previous stroke event.

“sex” should be replaced by “gender”.

Infections such as HIV, hepatitis and Covid-19 are conditions frequently associated with systemic inflammatory profiles, alike Chagas disease. These may contribute to behavioral and cognitive abnormalities, the presence of co-infections in CD patients and other disorders, here described as non-CD HF, should be described. Here or in the discussion section.

Infections such as HIV, hepatitis and Covid-19 are conditions frequently associated with systemic inflammatory profiles, alike Chagas disease. These conditions may contribute to behavioral and cognitive abnormalities, the presence of co-infections in patients with CD and other disorders, here described as non-CD HF, should be described. Here or in the discussion section.

Again, considering factors that may confound conclusions, the period of analysis should be described (previous, during, after Covid-19 pandemic?). Were similar numbers of patients (CD and non-CD) analyzed in these periods in groups (CD and non-CD)? Was long-Covid-19 excluded as a confounding factor?

Introduction:

Although the sequence of themes is well-presented, this section is superficially constructed.

Characteristics of chagasic cardiomyopathy are vaguely described, and data supporting frequency and traits of HF (with or without reduction of LVEF?) and embolic events in Chagas disease patients were not presented to readers to support the posed goal.

“Brain involvement in CD is usually attributed to the indirect effect of cardiomyopathy…”, citing references #8-10. However, the other side of this coin is not presented. Previous studies in Chagas disease have ruled out the contribution of cardiac dysfunction to brain involvement in other behavioral disorder (depression). Such data should be at least mentioned as an argument to evaluate if it is also true for cognitive impairment. Further, a recent preclinical study has approached this question, supporting dissociation of cardiac disease and behavioral (anxiety/depressive-like behavior) and cognitive changes in fluoxetine-treated mice. Such findings should be mentioned in introduction or at least discussed. Please, do not argue that “experimental data are not cited because preclinical models do not reproduce aspects of Chagas disease”. It is game over in science in general, and several publications support that some clinical aspects of chronic CD are well reproduced in mice, dogs and monkeys.

However, the other side of this coin is not presented. Previous studies on Chagas disease have ruled out the contribution of cardiac dysfunction to brain involvement in a behavioral disorder (depression). Such data should at least be mentioned as an argument to assess whether it is also true for cognitive impairment. Furthermore, a recent preclinical study has addressed this issue, supporting the dissociation of cardiac disease and behavioral (anxiety/depressive-like behavior) and cognitive alterations in mice treated with fluoxetine. Such findings should be mentioned in the introduction or at least discussed. Please do not argue that "experimental data are not cited because preclinical models do not reproduce aspects of Chagas disease". This is an outdated argument in science in general, and several publications support that some clinical aspects of chronic CD are well reproduced in mice, dogs and monkeys.

“However, authors could not exclude residual confounding from ventricular dysfunction itself or cerebrovascular disease. ” This sentence creates an expectative that in the present study a deep analysis of ventricular dysfunction and cerebrovascular disease, however it is not the case. Please, rephrase it.

The main goals of the study are clear: the present study: determine in HF patients if an association between CD and cognitive function is present when compared to non-CD patients; and to discriminate which cognitive domains are primarily affected in CD.” However, these goals are hermetic to public interested in Chagas disease, and most readers of PLoSNTD are not familiar to “cognitive domains” and how to approach them. Please, add information and references considering “cognitive domains” and the methods chosen to evaluate cognitive impairments in the present study. Considerer it is not a request for a revision of these themes, but the addition of some sentences with basic information is required.

Methods

Please, organize it in section such as (only suggestions of subtitles) “Study populations”, “Ethical statements (including inclusion/exclusion criteria)”, “Clinical characterization of cardiac disease (including traits and values for HF characterization)”, “Cognitive tests”, “Statistical analysis and treatment of data”.

“The study was approved by the local ethics committee…” Please, include the numbers of these licenses.

“We collected all information by consulting medical records and confirmed collected data using a standardized questionnaire.” Please, clarify: 1- nature of “all information” collected.; 2- EKG and ECHO data – some information how these assays were performed. For example, LVEF (Simpson %?); 3- “standardized questionnaire” (add to supplementary information).

"...patients with HF from four different outpatient clinics in two states of Brazil." Please. clarify: name of the Units and Brazilian states.

2020-2021-included patients: were they checked for Covid-19? Is there any bias of inclusion in CD or non-CD group in this period?

"...confirmed by an ELISA assay performed at a single central laboratory..." Please clarify: 1- by definition, diagnosis of Chagas disease requires two independent tests. Were the patients tested using two methods, as recommended? 2- name the "central laboratory".

Page 8: support with literature all used tests to evaluate cognition. Indicate and describe if any alteration of the original tests was done.

"...based on normalized values of the Brazilian population". Please, support by literature and more comprehensible information.

"…based on the USA population.". Please, support by literature and more comprehensible information.

Results

In general, data are very interesting and well-presented. However, some doubts require clarification.

Table 1 - Please, replace “race” by ethnicity.

“Chagasic group was older (57+11 vs 54 ±14 years), more likely to be female…” (page 14). These are two conditions associated with increased frequency of degenerative disease. May these aspects, independent of infection, explain the findings that “Chagas disease was associated with a lower score on MMSE when compared to non-CD patients” (page 15)?

“Chagas disease cardiomyopathy (CDC) and other cardiomyopathies (OC)” – It is confusing. What are the other cardiomyopathies (ischemic? Other infections?). Please, clarify it.

Considering NYHA classification, Table 1 shows that most of the patients are classified as groups I (without HF) and II (mild HF). All patients included in the study show signs of HF (included groups II- Mild? III- Moderate? IV severe?, excluding group I?)? In these cases, patient with HF were with and without reduced LVEF?

Data shown in table 2 referee to what groups of patients, regarding HF and LVEF%? 234 CD patients were studied, however 56/244 were classified in NYHA group I (without HF); 252 non-CD patients were included, however 51/259 were classified in NYHA group I. Please, clarify it. And modify the goals or the presentation of the results.

Further, there is a dispersion of LVEF% (means +/- SD, range), comparing CD and non-CD), is there a correlation between HF (and/or reduction of the LVEF%) and degree of mnemonic or cognitive impairment in each group (CD and non-CD)?

Page 11, “memory impairment did not remain associated with CD” – thus, in this cohort do you observed association between memory impairment and HF, independent of the etiology of the cardiomyopathy? These results could corroborate data of the literature and add value to this work studying CD patients.

The goals of this study were “determine in HF patients if an association between CD and cognitive function is present when compared to non-CD patients; and to discriminate which cognitive domains are primarily affected in CD.” If I understood it correctly, these are two independent questions. To answer question 1- groups of study should include CD and non-CD patients with HF (discriminated as mild, moderate, severe). To answer question 2 -all included patients (discriminating – NYHA group without HF - group I - and groups with HF), controls are all patients in the non-CD group. If question 2 is also focused on HF patients, it should be mentioned. In other words, it is necessary to focus the data analysis on the 2 questions posted or modify these questions.

Please, clarify all above presented points considering that the main goal of this study was “determine in HF patients if an association between CD and cognitive function is present when compared to non-CD patients; and to discriminate which cognitive domains are primarily affected in CD.”

Representative figures of some mnemonic tests (such as original figures and patients draws or similar representative data) may help readers to follow the findings. One should consider that there are a reduced number of articles evaluating mnemonic traits in Chagas disease patients. Readers interested in Chagas disease are not used to these tests. Education of readers is a relevant aspect of a scientific publication.

The literature shows: 1- Electrical abnormalities are frequently associated with inflammatory cardiomyopathies (eg, Lazzerini et al., 2023). 2- Inflammatory profile is associated with behavioral/cognitive alterations (eg, hepatitis, Covid-19). Thus, considering that the frequencies of patients with rhythm and conduction abnormalities are higher in CD (vs non-CD), may these changes be related with memory/cognitive abnormalities in CD group?

Use of alcohol may contribute to liver/brain axis alterations, including cognitive changes. Comparison of “Alcohol use” among CD and non-CD shows difference (p = 0.007), and more non-CD registered “often-past” use of alcohol. Should it be also considered as a confounding factor in the present study? Please, consider in a revision of the results.

Discussion

Globally, discussion is well presented. Some points may be better explored.

“large cohort of HF patients”. In fact, what were the numbers of patients with HF included in the groups (CD vs non-CD). Please, clarity it in the Results section.

“The mechanisms underlying the pattern of CD-associated cognitive impairment are under investigation in another study by our group (reference #11).” The argument is not clear as the cited reference is Oliveira-Filho et al. 2012.

“Inflammatory biomarkers, seen in Chagas disease (reference #18), have been associated with brain atrophy in a cohort of 1926 individuals (reference #19)”. Please, rephrase this sentence. It may mislead readers. Article #19 studies Alzheimer disease patients. More than brain atrophy, systemic and local brain inflammations have been associated with behavioral/cognitive changes. Preclinical studies in a model of chronic CD may add important insights to this discussion.

“We did not include a normal control group because we wanted to control for the presence of cardiac disease and investigate the isolated effect of CD on cognition.” I agree, that “normal control” (term that should be replaced by group of patients seronegative for CD or some similar term) is not a point in the present study as the goals of this study were “determine in HF patients if an association between CD and cognitive function is present when compared to non-CD patients; and to discriminate which cognitive domains are primarily affected in CD.” Again, these are two independent questions, as previously argued. As data are presented in a confusing way, confusing arguments may add more confusion.

“A neuroimaging component of the present study is ongoing and should help answering which mechanisms are implicated in CD-associated cognitive impairment.” The present study included patients in the period 2007-2021. Previous publications of this group showed brain atrophy in CD patients (reference #4, published in 2009). Were some of these previously studied patients included in the present study? These findings on brain atrophy may add crucial information to the present MS.

Usually, illiteracy/incomplete elementary school (as data shown in Table 1) is linked to poverty, and poor people have reduced access to mental health care. If any, what was the feedback to patients, particularly to the ones with cognitive impairments and CD? Some words on these may improve discussion.

Poverty may trigger/aggravate mental disorders (eg, Ridley et al., 2020), thus some words on this topic may improve discussion.

Limitations of the study were presented.

Reviewer #2: I think the manuscript is important and deserves publication at PLoS NTD. Yet, my major concern is that the two groups may differ in several relevant information, such as those described above with the chagasic group being formed by people who live or lived at the country side, with a life pattern simple (maybe without formal education) but in a less stressing situation, while those from the non-chagasic group may have lived in a more competitive, more incertaint milieu that could lead to higher stress, a situation that could lead to higher stress.

PLOS authors have the option to publish the peer review history of their article (what does this mean? ). If published, this will include your full peer review and any attached files.

**Do you want your identity to be public for this peer review?** For information about this choice, including consent withdrawal, please see our Privacy Policy .

Reviewer #1: No

Reviewer #2: No

---

## [Editor Report · Decision Letter 1]

12 Mar 2025

Dear Sra Serrano,

We are pleased to inform you that your manuscript 'COGNITIVE IMPAIRMENT IN CHAGAS DISEASE PATIENTS IN BRAZIL, 2007-2021: A CROSS-SECTIONAL STUDY' has been provisionally accepted for publication in PLOS Neglected Tropical Diseases.

Best regards,

Hira L Nakhasi, Ph.D.

Section Editor

Hira Nakhasi

Section Editor

Shaden Kamhawi

co-Editor-in-Chief

Paul Brindley

co-Editor-in-Chief

The authors have satisfactorily responded to reviewers comments.

---

## [Editor Report · Acceptance letter]

Dear Sra Serrano,

We are delighted to inform you that your manuscript, "Cognitive Impairment in Chagas Disease Patients in Brazil, 2007-2021: A Cross-Sectional Study ," has been formally accepted for publication in PLOS Neglected Tropical Diseases.

Best regards,

Shaden Kamhawi

co-Editor-in-Chief

Paul Brindley

co-Editor-in-Chief
